# Ferroptosis: A New Development Trend in Periodontitis

**DOI:** 10.3390/cells11213349

**Published:** 2022-10-24

**Authors:** Kexiao Chen, Shuyuan Ma, Jianwen Deng, Xinrong Jiang, Fengyu Ma, Zejian Li

**Affiliations:** 1Medical Center of Stomatology, The First Affiliated Hospital, Jinan University, Guangzhou 510630, China; 2School of Stomatology, Jinan University, Guangzhou 510630, China; 3Chaoshan Hospital, The First Affiliated Hospital of Jinan University, Chaozhou 515600, China

**Keywords:** ferroptosis, periodontitis, iron overload, lipid peroxidation, oxidative stress, periodontal pathogen

## Abstract

Periodontitis is a chronic inflammatory disease associated with bacterial biofilm. It is characterized by loss of periodontal support tissue and has long been considered as a “silent disease”. Because it is difficult to prevent and has a health impact that can not be ignored, researchers have been focusing on a mechanism-based treatment model. Ferroptosis is an iron-dependent regulatory form of cell death, that directly or indirectly affects glutathione peroxidase through different signaling pathways, resulting in a decrease in cell antioxidant capacity, accumulation of reactive oxygen species and lipid peroxidation, which cause oxidative cell death and tissue damage. Recently, some studies have proven that iron overload, oxidative stress, and lipid peroxidation exist in the process of periodontitis. Based on this, this article reviews the relationship between periodontitis and ferroptosis, in order to provide a theoretical reference for future research on the prevention and treatment of periodontal disease.

## 1. Introduction

Ferroptosis is a concept proposed by Dixon et al. [1] in 2012, and is used to describe a regulatory cell death phenomenon induced by the small molecule erastin that is different from autophagy, apoptosis, and necrosis. In 2018, the Nomenclature Committee on Cell Death (NCCD) defined ferroptosis as the oxidative response of the intracellular microenvironment regulated by glutathione peroxidase 4 (GPX4), a form of regulatory cell death triggered by stimulation that can be inhibited by iron chelators and lipophilic antioxidants [2]. Recent studies [3,4,5] have found associations between iron-induced cell death and a variety of diseases, including vascular diseases, tumors, neurodegenerative diseases, osteoporosis and diabetes.

According to previous studies, iron (Fe), as an active metal, actively participates in the redox cycle reactions, and has the ability to produce active free radicals in the Fenton reaction [6]. This mediated by an unbalanced state of oxidative stress will ensue lipid peroxidation, which is an important link to ferroptosis. During the maintenance of iron homeostasis, hepcidin plays a central role, and its expression is regulated by iron concentration, hypoxia, and inflammation [7,8]. Several studies [9,10,11] have confirmed that the iron balance is indeed closely related to the inflammatory response.

Periodontitis, as a common bacterial inflammatory reaction in the oral cavity, is the main cause of tooth loss in elders and has a great influence on their oral health and daily life. Scholars have successively proven that the development of periodontitis is accompanied by iron overload [12,13] and glutathione (GSH) depletion [14,15]. Based on this, we hypothesize that periodontitis is related to oxidative stress and ferroptosis. This article will systematically review the related mechanisms of ferroptosis and its correlation with periodontitis to provide a foundation for relevant research on the pathogenesis of periodontitis in the future.

This review mainly focuses on the period 2010–2022. The main focus of this article is the basic medical research on ferroptosis and periodontitis. Research included in this paper were filtered through keyword searches in PubMed, MEDLINE, Google Scholar, ProQuest, and Web of Science databases. These databases are searched when available using a combination of subject terms and filters, such as time. We didn’t impose any language restrictions. Keywords included “oxidative stress”, “periodontitis”, “inflammation”, “lipid metabolism”, “iron overload”, “ferroptosis” and “periodontal microbiome”. We reviewed, integrated, and summarized the retrieved literature to systematically review the current understanding of ferroptosis, focusing on their underlying mechanisms, associations, and regulation, discussing their role in periodontitis, and considering emerging therapeutic opportunities and challenges associated with future prospects.

## 2. Ferroptosis

### 2.1. Biological Function of Iron

Iron is an important element for organisms to carry out biological functions normally [16]. Iron participates in blood circulation in the form of transferrin (Tf), binds to transferrin receptor 1 (TFR1) on the cell membrane to form a complex, and is then endocytosed by the cell. The internal acidification of the endocytosed inclusion bodies releases the reduced ferrous Ion (Fe^2+^) from Tf and transports it to the cytoplasm through divalent metal transporter 1 (DMT1, also known as SLC11A2). The complex then returns to the cell membrane to participate again in this circulation [17]. When the body is healthy, it is generally in a state of iron balance, where excess iron exists in the form of ferritin, and free iron is not at an excessive level [18]. Instead, iron overload can occur in pathological conditions such as genetic mutations (most commonly hereditary hemochromatosis), surgery, excessive dietary iron intake, or chronic blood transfusion therapy [19,20]. The two different forms of iron correspond to different functions. Among them, ferritin is a kind of protein that regulates intracellular iron steady-state. It is wrapped in a hollow heterogeneous body wrapped in a ferritin heavy chain and ferritin light chain. This capability makes it dual function of storing and detoxifying iron [21,22]. Free iron is highly reactive and is involved in DNA synthesis, oxygen transport, and ATP production, which are essential for maintaining the biological function of proteins [23,24]. Once intracellular iron accumulates, resulting in iron overload, it can trigger the Fenton reaction and catalyze the production of large amounts of reactive oxygen species (ROS), leading to cell death and overall oxidative damage to tissues [25,26].

### 2.2. Characteristics of Ferroptosis

Ferroptosis is accompanied by cellular metabolic disorders, including iron metabolism, lipid metabolism, and amino acid metabolism. It shows the following characteristics: (1) Morphology [27]: The nuclei are normal in size, and there is no condensation of chromatin; the mitochondria shrink, and the mitochondrial cristae blur or disappear; the mitochondrial membrane density increases, and cell membranes and mitochondrial membranes are broken; (2) Biochemical metabolism [26,28]: The redox homeostasis of the body cells is disrupted; the levels of ROS and its derived free radicals in tissues are increased, accompanied by GPX4 consumption, accumulation of lipid peroxidation, increased oxidation of nicotinamide adenine dinucleotide phosphate (NADPH), activation of mitogen-activated protein kinase (MAPK) signal transduction pathways, and a decline in antioxidant ability; and (3) Immunology [2]: Injury-related molecular patterns are activated, proinflammatory mediators are released, and inflammation is induced or aggravated.

### 2.3. Pathways of Ferroptosis

#### 2.3.1. Induction of Ferroptosis by Inhibiting the Cystine/Glutamate Transport Receptor (System X_c_^−^)

System X_c_^−^ is a reverse transporter on the cell membrane consisting of two subunits, SLC7A11 and SLC3A2. Glutamate is transferred out of the cell, and cystine is transferred into the cell at a ratio of 1:1. Cystine is reduced to cysteine and participates in the synthesis of GSH [28]. GPX4 is an important enzyme to maintain the steady state of oxidation and reduction in the body, and GSH is a cofactor for its activation. When the level of GSH decreases, the activity of GPX4 decreases, lipid peroxidation occurs in cells, ROS levels rise and accumulate, and the ferroptosis process is initiated [26,29,30,31]. For this reason, GPX4 is regarded as a crucial target of ferroptosis [31].

#### 2.3.2. Induction of Ferroptosis by Voltage-Dependent Anion Channel (VDAC)

VDAC, a channel protein found widespread in the mitochondrial outer membrane, maintains its permeability by transporting iron ions and metabolic waste [32]. Mitochondria are the main sites for iron metabolism in cells. Iron is involved not only in energy synthesis but is also considered as an important auxiliary factor of the electron transfer chain and metalloproteinases in mitochondria. An imbalance in the “on-off” status of VDAC affects the metabolism of the cell. When the channel is open, respiratory substrates enter, mitochondrial metabolism is active, free radical production increases and is converted into hydrogen peroxide by superoxide dismutase (SOD) in the mitochondrial matrix, and hydroxyl radicals with high activity are formed by the Fenton reaction. ROS causes mitochondrial dysfunction through the MAPK pathway, and oxidative stress in cells induces ferroptosis [33,34].

#### 2.3.3. Induction of Ferroptosis by Ferritin Phagocytosis

In general, iron is in equilibrium within the cell, with most of it stored in the form of ferritin. Ferritin, a common ferri-binding protein, has the properties of nontoxic chelated iron, which can limit the generation of free radicals triggered by inflammation. Ferritin can be delivered to autophagosomes by nuclear receptor coactivator 4 (NCOA4), which degrades and releases free iron in autophagosomes. Free iron is mediated to form an iron pool in the cytoplasm, and the Fenton reaction occurs, which leads to an increase in ROS levels and finally ferroptosis [35,36]. Excessive autophagy causes ferroptosis, which is consistent with the research results of Zhou [37], Bai [38] and Yang [39].

#### 2.3.4. Induction of Ferroptosis by the P53 Pathway

Ferroptosis is closely related to abnormal intracellular metabolism, and the P53 plays an important role in regulating cell metabolism [40]. P53 inhibits the expression of SLC7A11, which decreases the function of system X_c_^−^, sensitizing the cells to ferroptosis [41]. The target genes of P53 also include glutaminase 2 (GLS2) and prostaglandin-endoperoxide synthase 2 (PTGS2). PTGS2, a pivotal enzyme for the compound prostaglandin (PG) in the body, can regulate the level of intracellular lipid peroxide and change the sensitivity to ferroptosis. Thus, PTGS2 is regarded as a marker of ferroptosis [30]. Gao et al. [42] found that GLS2 can catalyze the hydrolysis of glutamine to glutamate and reduce GSH levels. An increase in GLS2 levels and a decrease in GSH levels are necessary conditions for ferroptosis. Polyunsaturated fatty acids (PUFAs) are increased and esterified to CoA (CoA) by acyl-Coa synthase long-chain family member 4 (ACSL4). Lysophosphatidyl choline acyltransferase 3 (LPCAT3) integrates it into cell membranes and becomes a substrate for lipid oxidation driven by lipoxygenase (LOX) [43,44,45]. In this process, 5-LOX is a direct target gene of P53, which mediates key lipid peroxidation during iron ptosis [46,47]. The above evidence suggests that P53 regulates and controls ferroptosis through other genes and not directly.

However, the regulatory network of P53 in ferroptosis is enormous and intricate. Research has shown that P53 reduces ferroptosis sensitivity by inhibiting the activity of dipeptidyl peptidase-4 (DDP-4) in some cases [48]. Tarangelo et al. [49] used nutlin-3 to inhibit the activity of ubiquitin ligase murine double minute 2 so that P53 could exist stably without being subject to ubiquitin degradation. These authors found that the ferroptosis process in cells pretreated with nutlin-3 was suppressed, which indicates that stably existing P53 can delay the occurrence of ferroptosis in cells.

#### 2.3.5. Induction of Ferroptosis by the MAPK Pathway

Erastin-induced ferroptosis is often accompanied by the activation of the MAPK pathway, indicating that there may be a close relationship between the activation of the MAPK pathway and ferroptosis. Apoptosis signal-regulating kinase-1 (ASK1), as an upstream kinase of P38, has been shown to cascade with corresponding oxidative stress [50]. A Japanese study [51] found that persistent and severe cold stress may induce activation of the ASK1-P38 axis through ROS signaling, leading to cell ferroptosis, which can be significantly reduced after the administration of the iron chelating agent deferoxamine, further confirming the association between ferroptosis and the P38/MAPK channel. It was found that the expression of MAPK-related proteins PJNK1/2 and PERK1/2 was increased in the model of arsenic-induced ferroptosis but only in the higher concentration of arsenate, and the increase was not dose-dependent [52]. Although the specific mechanism is unclear, this result is similar to another study [27] that prevented erastin-induced ferroptosis by blocking the expression of the MAPK channel protein. Lv [53] treated human osteosarcoma cells with β-Phenethyl isothiocyanate (PEITC) and found that unstable iron levels increased within the cells, causing GSH depletion, which resulted in oxidative stress and lipid peroxidation, and observed that the ERK, P38, and JNK pathway in cells were significantly activated, confirming that PEITC treatment caused the MAPK signaling pathway activation on which ROS depends. However, the experiment indicated that PEITC not only causes ferroptosis but also apoptosis, autophagy, and other forms of cell death by inducing oxidative stress. The crosstalk and mutual relationships between different death paths are still elusive and need further research.

#### 2.3.6. Inhibiting Ferroptosis by the Hippo Pathway

The Hippo pathway is mainly composed of mammalian Ste20-like kinases 1/2 (MST1/2), large tumor suppressor 1/2 (LATS1/2), regulatory molecule Salvador 1 (SAV1), and transcriptional coactivator Yes-associated protein (YAP). When the cell density increases, the E-calcium-adhesive protein increases cell-to-cell bonding, and the cell density information is transmitted to the Hippo pathway through neurofibromatosis 2 (NF2), which mediates phosphorylation of LATS1/2 while reducing the ubiquitin degradation of LATS1/2 [54], thus enhancing the activity of LATS1/2 and reducing the sensitivity to ferroptosis. This phenomenon may be due to the phosphorylation of YAP by active LATS1/2, thus, the YAP-TEAD (TEA domain) -targeted downstream geneACSL4 and transferrin receptor (TFRC) expression was inhibited [55]. In addition, there is evidence that the Hippo pathway has a mutual interaction with the MAPK pathway [56].

#### 2.3.7. Inhibiting Ferroptosis by Nuclear Factor Erythroid 2-Related Factor 2 (NRF2)

Nuclear factor erythroid 2-related factor 2 (NRF2) is an important transcription factor in the oxidative stress response [57]. The target genes are negative regulatory factors of ferroptosis, such as quinone oxidoreductase 1 (NQO1), heme oxygenase 1 (HO-1), and ferritin heavy chain, which play key roles in resisting ferroptosis [58]. Under normal conditions, one of the domains of NRF2 (Neh2) binds to the DGR (double glycine/Keclch repeats) region of its negative regulatory factor Keapl is degraded by ubiquitination in the cytoplasm and maintained at a relatively stable concentration. However, under oxidative stress conditions, Keapl is decoupled from NRF2, which binds to antioxidant reaction elements (AREs) to regulate the activity of target genes, such as superoxide dismutase and catalase and to initiate the expression of II-phase detoxifying enzymes and GPX4 to remove excessive amounts of harmful substances, such as ROS [59]. After the treatment of hypoxic-ischemic brain injury in mice with melatonin, inhibition of neuronal ferroptosis is found [60]. On this basis, after treatment with an NRF2 inhibitor, the expression of NRF2 and GPX4 is downregulated, the expression of lipid peroxide is enhanced, and the protective effect of melatonin on neurons is weakened, indicating that the NRF2-GPX4 signaling pathway might inhibit the occurrence of ferroptosis.

HO-1, one of the II phase detoxifying enzymes that degrade haem to produce biliverdin/bilirubin, carbon monoxide (CO), and Fe^2+^ and regulate intracellular oxidation and iron metabolism. Although bilirubin, biliverdin and CO play an important role in antioxidant stress, the amount of intracellular iron and ROS are the leading factors in the process of ferroptosis, and excessive iron and ROS may reverse the protective effect of HO-1 [61]. This may also explain why the results of Adedoyin [62] and Kwon [63] ran counter to two experimental results. At the same time, other studies have confirmed this dual effect of HO-1. HO-1 is significantly induced in inflammatory tissues, whereas ferritin acts as a recognized inflammatory marker, and the heavy chain of one of its components mediates the protective effect of HO-1 against oxidative stress [64]. However, in NRF2/HO-1 signaling, overexpression of HO-1 causes the accumulation of unstable intracellular iron pools by upregulating TF, TFRC, FTH1, and NCOA4 expression, and subsequently triggers cell ferroptosis [65]. Various lines of evidence suggest that HO-1 may be an essential enzyme for iron death, and that the NRF2/HO-1 axis plays an important role between ferroptosis and antioxidative stress and iron accumulation [66].

## 3. Ferroptosis and Periodontitis

At present, the specific mechanism of the development of periodontitis, a multifactor disease, is not clear. Certainly, periodontitis is an infectious inflammatory disease caused by the interaction of micro-organisms in dental plaque biofilms with the host, environment, and other factors. Iron-dependent oxidative stress and lipid peroxidation are thought to be common mechanisms of ferroptosis and inflammatory disease. Periodontal tissue in the infected state shows up-regulation of iron-containing compounds such as ferritin, transferrin and heme [67,68,69]. Periodontal pathogens chelate iron from these compounds for their own growth and reproduction [70,71]. *Porphyromonas gingivalis*, for example, encodes a variety of proteins that bind to red blood cells. When heme is not readily available, it can also be released by lysing red blood cells, this increases the concentration of iron in the cells [72]. Excess free iron occurs Fenton reaction, catalyzing the production of free radicals, which brings ROS to the body. The periodontal pocket presents an anoxic environment due to the selection and prevalence of subgingival anaerobic bacteria. Hypoxia-inducible factor-1 (HIF-1) is an adaptive response to hypoxia, and it has been confirmed by many studies that it is highly expressed in periodontal inflammatory tissues [73,74,75]. It has been reported that HIF is a master regulator of many genes related to iron homeostasis and inflammation, including TFR1, HO-1, DMT1, erythropoietin, and ferritin [10], while promoting fatty acid deposition in the microenvironment [76]. PUFA is closely related to the occurrence of periodontitis and has been confirmed by lipidomics to be one of the lipids most prone to peroxidation in ferroptosis [44,77]. In ROS-induced periodontal oxidative stress, the double bond in PUFA molecular structure is the preferred site of oxidation, and therefore prone to produce lipid peroxidation in cells [78]. 

In addition to the ROS produced by free iron, in order to resist microbial infection, the body’s immune defense cells, mainly pleomorphic leukocytes and polymorphonuclear leukocytes, also catalyze “respiratory burst” through NADPH oxidase (NOX), consume NADPH and produce a large number of ROS [79,80]. Overwhelming evidence shows that ROS causes lipid peroxidation, protein damage, and nucleic acid damage in the body [81]. Long-term high levels of ROS may deplete the antioxidant pool in the cysteine-GSH-GPX4 pathway and promote the imbalance of the oral oxidation-antioxidant system. This oxidative stress undoubtedly exacerbates inflammation. It’s a complex feedback. We can understand that the feed-forward mechanism of “microbial infection-ROS-inflammation” seems to accelerate the susceptibility of the body to infection, inflammation and oxidative stress in turn.

There are studies [82] showing that oxidative stress and lipid peroxidation are closely related to periodontitis. There are changes in GPX, the ratio of glutathione/oxidized glutathione, and myeloperoxidase (MPO) activity, increases in lipid peroxides, and other indicators in the tissues of periodontitis patients, and GPX elevation may be an antioxidant compensation during oxidative stress. There are many similar indicator changes between periodontitis and iron-induced cell death, indicating that there may be some correlation between the two processes.

### 3.1. Iron Overload

Physiologically, there is less iron in free form. Changes in iron concentrations have also been shown to correlate with the progression of active periodontitis and the severity of periodontitis [12,83]. Iron overload may be one of the trigger factors promoting dysbiosis of periodontal tissues, inflammation and oxidative stress [84]. Studies have shown that hereditary hemochromatosis, characterized by systemic iron overload, promotes the development of severe periodontitis [85]. Both animal experiments and clinical experiments have shown that the iron content in periodontitis sites was higher than that in healthy periodontal sites [68]. It has been reported that free iron concentrations are significantly up-regulated 6–12 h after periodontal infection [86], and these high concentrations of iron have been shown to be more favorable for the maturation of certain periodontal-dominant pathogenic bacteria biofilms [87]. To satisfy their growth and reproduction, micro-organisms need to compete with their hosts for iron. Periodontal pathogens, such as *Porphyromonas gingivalis* and *Prevotella intermedia*, can decompose and utilize iron-binding proteins in the host to provide an iron source for their metabolism. Ferritin fragments or free iron are generated when iron-binding proteins are cleaved, resulting in a local iron-overload environment [88,89,90]. Iron overload promotes ROS production in periodontal tissue and enhances the growth and virulence of periodontal pathogens [68], ultimately inducing cellular ferroptosis and periodontal tissue destruction [91]. These studies suggest that iron metabolism is altered in the context of periodontitis and then feeds back into the development of periodontal inflammation.

It is worth noting that the state of “iron overload” in periodontal tissue is accompanied by a decrease in circulating iron and an increase in the negative iron-regulating hormone (hepcidin) [92]. Typically, elevated circulating iron levels induce hepcidin production, which subsequently triggers the endocytosis and degradation of ferroportin to reduce the entry of iron into the circulation from cells such as macrophages. Conversely, hepcidin promotes cellular iron into circulation [93]. However, inflammatory mediators, such as interleukin-6 (IL-6) and bacterial lipopolysaccharides, produced upon microbial stimulation interfere with the regulation of iron homeostasis by hepcidin, further exacerbating intracellular iron retention and circulating iron levels decrease [94]. This was further supported in studies related to periodontitis [95,96]. Therefore, we speculate that although the body has made a series of defensive changes to the “iron overload” state of periodontitis, these changes may exacerbate the vicious circle of “iron overload” to a certain extent under the stimulation of periodontal infection.

### 3.2. Lipid Peroxidation

The content of lipid peroxides in saliva and the gingival crevicular fluid of patients with chronic periodontitis is increased [97]. The metabolism of periodontal pathogenic bacteria is abnormal. Neutrophils aggregate and produce phospholipase, which generates arachidonic acid (AA) under the stimulation of inflammatory mediators and then metabolizes prostaglandin and leukotriene to aggravate periodontal bone absorption. Compared with the gingivitis group and the periodontal healthy group, the expression of metal ions such as sodium, magnesium, potassium, calcium, iron and selenium was significantly increased in the periodontitis group [98]. These metal ions have been shown to play key roles in modulating the immune-inflammatory process of periodontitis as well as redox reactions [99,100]. Among them, iron ion is one of the main players in the redox reaction, which can lead to the massive production of ROS, which can cause oxidative damage to the lipids and other components in the periodontal tissue, thereby promoting the progression of periodontitis.

A significant increase in MDA, a by-product of lipid peroxidation, and a marker of ferroptosis, has been reported in samples from patients with periodontitis [101], which corresponds to the observed trend of decreasing ferric ion reducing antioxidant power [102]. In addition, stimulated by the ferroptosis activator ^®^erastin, the expression levels of GSH in periodontal pathogens and periodontal ligament stem cells were significantly decreased, while the levels of ROS, Fe^2+^ and MDA were significantly increased. However, the expression of MDA was greatly reduced after treatment with the ferroptosis inhibitor Ferrostatin-1 [103]. It can be seen that the occurrence of lipid peroxidation in periodontal tissue is closely related to ferroptosis. In periodontal inflammatory tissue, PUFAs increase and undergo a series of peroxidation reactions driven by upregulated iron and LOX [104,105].

### 3.3. Ferritin Phagocytosis

In periodontitis, the flora of periodontal biofilms is out of balance, and the content of anaerobic bacteria increases, such as *Porphyromonas gingivalis*, *Actinobacillus actinomycetemcomitans*, and *Fusobacterium nucleatum*, which can produce a large number of short-chain fatty acids (SCFAs), including lactic acid, acetic acid, and butyrate, to regulate the inflammatory reaction [106]. Studies [107] have shown that the concentration of SCFAs in the gingival crevicular fluid of periodontitis patients is positively correlated with the depth of the periodontal pocket. Butyrate, as an important short-chain fatty acid in the periodontal pocket, can inhibit the growth and proliferation of periodontal ligament fibroblasts (PDLFs), induce the expression of NCOA4 in PDLFs, and stimulate ferritin phagocytosis. In addition, with the increase in butyrate levels, iron and ROS accumulation, GSH and GPX consumption, ACSL4 expression enhancement and lipid peroxidation occur, resulting in ferroptosis [108]. Through the triggers outlined above that initiate and promote the ferroptosis process, the formation of a high ROS level and lipid peroxidation of the microenvironment in the face of changes in the extracellular environment, periodontal stability is destroyed, the skin barrier of the gums is damaged and the periodontal ecology leads to cellular inflammation, inflammatory aggravation, or even death.

### 3.4. MAPK Pathway

The MAPK pathway is closely related to cell proliferation, differentiation, inflammation, apoptosis, and other cellular reactions and physiological and pathological processes, especially the ERK, JNK, and P38 proteins, which are sensitive to intracellular oxidative stress [109,110]. Inflammatory sites are often infiltrated by T cells, which have the ability to synthesize and secrete ferritin [111]. Huang et al. [112] found that the content of ferritin in the gingival crevicular fluid of periodontitis patients is higher than that of healthy people and is related to the degree of local inflammatory infiltration. Ferritin induces the production of various inflammatory factors such as interleukins and tumor necrosis factor, and this process occurs by inducing the ERK/P38 MAPK pathway. At the same time, it was found that bacterial endotoxin (lipopolysaccharide) and inflammatory factors can increase ferritin levels, and inhibition of the ERK and P38 pathways can reduce the proinflammatory effect of ferritin. HIF in the local microenvironment of periodontitis is itself controlled by the transcription of the major immunomodulator NF-kB and is also a major regulator of many genes involved in iron homeostasis and inflammation, including TFR1, HO-1, DMT1, erythropoietin, and ferritin [113,114]. In addition, increased ferritin concentrations in gingival crevicular fluid have been observed in patients with periodontitis [115,116]. Stimulation of preosteoclasts with it can enhance the phosphorylation of JNK, P38 and other MAPK-related channels by increasing intracellular iron content. However, treatment with iron chelators inhibited their activation and the expression of downstream molecules, and even inhibited the activation of MAPKs in the presence of ROS, thus inhibiting osteoclast generation. This may be related to the induction of HO-1 by iron chelators [117]. Periodontal inflammation can up-regulate ferric-ion binding protein level and aggravate iron load, and ferritin can induce the expression of inflammation-related MAPK channels and aggravate inflammatory response. This feedback forms a cascade amplification effect, causing a vicious circle and aggravating periodontal damage.

### 3.5. P53 Pathway

The P53 signaling pathway is a huge, delicate, and complex regulatory system. Research [118] has shown that the upregulation of P53 expression in patients with periodontitis is consistent with the progression of periodontitis because the hypoxia and inflammatory environment of periodontitis induce P53 expression, and hypoxia and lipopolysaccharide promote each other at the protein level. To a certain extent, P53 can regulate the expression of ROS-producing genes, such as inhibiting the expression of SLC7A11, one of the constituent subunits of system X_c_^−^, and thus increase the level of ROS in cells. However, under mild stress, P53 can promote the expression of antioxidants [118,119,120]. SAT1 gene is a transcriptional target of P53. Interestingly, its expression can induce lipid peroxidation and lead cells to ferroptosis due to oxidative stress. This function may be related to the induced expression of arachidonic acid and 15-LOX [121]. In addition to lipid peroxidation, LOX-mediated production of proinflammatory lipid mediators such as leukotrienes and prostaglandins is also an important factor aggravating periodontal inflammation [99,122]. PTGS2, one of the target genes of P53, is a rate-limiting enzyme for the synthesis of prostaglandins, which is an important inflammatory mediator leading to periodontal destruction and plays an important role in the progression of periodontitis.

### 3.6. Transforming Growth Factor β

Transforming growth factor-β (TGF-β) is an important substance in the process of periodontal tissue injury, and one subtype, TGF-β1, is an important factor in regulating the maturation and proliferation of periodontal ligament stem cells. In the osteogenesis of periodontitis, TGF-β1 can inhibit the expression of IL and promote the expression of osteocalcin. Recent research [123] showed that TGF-β1 inhibits the expression of SLC7A11 by activating Small mothers against decapentaplegic 3 (Smad3), which causes an oxidation-reduction imbalance and induces lipid peroxidation in cells. This finding suggests that vital osteogenic substances may participate in ferroptosis. However, the mechanistic relationship between periodontitis and ferroptosis is complex, and this specific relationship needs further study.

## 4. Clinical Application and Conclusions

Studies have shown that different degrees of iron metabolism disorder [124] and lipid peroxide accumulation [125] are found in periodontitis with ferroptosis characteristics, and regulation of ferroptosis can affect the process of periodontitis. In view of the fact that there are many common pathways between periodontitis and ferroptosis and many similar expression patterns of inflammatory factors and proteins, we reasonably hypothesize that ferroptosis may exist and participate in the occurrence and development of periodontitis (Figure 1 and Figure 2).

Studies of the association between ferroptosis and periodontitis are ongoing. Various drugs such as iron chelators and antioxidants have been attempted to block the onset of ferroptosis. The use of an antioxidant enzyme, Peroxiredoxin 6, has been reported to reduce lipopolysaccharide-induced inflammation and ferroptosis in periodontitis [126]. In addition, some studies have found that the use of iron chelators such as deferoxamine to bind free iron ions can effectively inhibit the growth of periodontal pathogens and the negative effect of iron overload on bone resorption in periodontitis [127]. Studies have shown that tetracycline antibiotic, as a classical drug for the treatment of adolescent periodontitis, seems to have a unique property—a strong iron-chelating activity [128]. This chelating property is related to the ability of tetracyclines to inhibit matrix metalloproteinase (MMP) activity. In fact, its iron-chelating function partly prevents MMP activation and ROS production that induce tissue breakdown. To the satisfaction of scholars, tetracycline complexation with iron not only does not affect the efficacy of tetracycline, but can also effectively reduce iron as an infection enhancer to tissue damage. This suggests that the reasonable inhibition of excessive iron has some application value in the treatment of periodontal disease.

A bioinformatic analysis provides some evidence for the association between fer-roptosis and periodontitis, indicating a significant association between ferroptosi-related genes and periodontitis, which is of good value for the diagnosis of periodontitis [129]. However, the specific mechanism, pathway, and key executive molecules between ferroptosis and periodontitis have not yet been determined. There are still many questions to be solved, such as the following: (1) Does ferroptosis also exist in physiological processes? (2) As ROS, iron, GPX, and other substances in periodontal disease are also related to autophagy [130] and apoptosis [131] does ferroptosis coexist with other forms of cell death in the pathogenesis of periodontitis, and which form is dominant in the disease process? (3) To what extent is lipid peroxidation related to ferroptosis? and (4) What are the characteristic markers of ferroptosis in periodontal tissues? Then, we can further understand how ferroptosis is initiated and regulated in the development of periodontitis by verifying whether ferroptosis exists in periodontitis. On this basis, antioxidation and anti-iron overload treatment may be considered in the prevention and treatment of periodontitis, providing a development direction and intervention target for the prevention and treatment of periodontitis.

## Figures and Tables

**Figure 1 cells-11-03349-f001:**
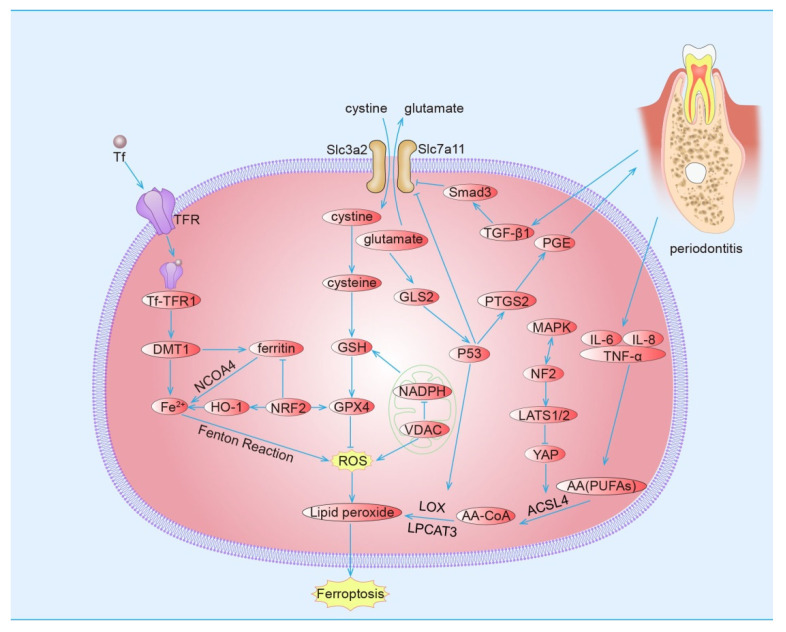
**The related mechanism of ferroptosis and periodontitis.** Glutamate acid and cystine are exchanged in a ratio of 1:1 inside and outside cells, and cystine is reduced to participate in GSH synthesis, activate GPX4, and inhibit ROS production. When VDAC was opened, NADPH production increased, which provided reduced hydrogen for GSH formation but increased ROS production. Tf combines with Tfr and merges into cells, and is introduced into the cytoplasm by DMT1, and exists in two forms of ferritin and free iron. Ferritin autophagy occurs through NCOA4, which increases the content of free iron, causes Fenton reaction and increases ROS. Under oxidative stress, NRF2 initiates the expression of GPX4, HO-1 and ferritin heavy chain, but at the same time, HO-1 can decompose heme to produce ferrous iron. P53 not only inhibits the expression of SLC7A11 but can also target PTGS2, promote prostaglandin synthesis, and damage periodontium. Glutamate was prepared by hydrolysis of glutamine catalyzed by GLS2. LOX promotes the formation of lipid peroxidation in cells. MAPK pathway and hippo pathway promote each other, NF2 activates hippo, activates LATS1/2, phosphorylates YAP and inhibits the expression of downstream ACSL4. In periodontitis, inflammatory factors IL-6, IL-8 and TNF-α can promote the deposition of unsaturated fatty acids, which are acylated by AcSL4, integrated to cell membrane by LPCAT3, and become an important substance of iron death after being oxidized by LOX. TGF-β1 during periodontal bone formation can inhibit the expression of SLC7A11 through Smad3. Abbreviations—Tf, transferrin; TFR, transferrin receptor; DMT1, divalent metal ion transporter 1; NCOA4, nuclear receptor coactivator 4; OH-1, heme oxygenase-1; NRF2, nuclear factor erythroid 2-related factor 2; GSH, glutathione; GPX4, glutathione peroxidase 4; ROS, reactive oxygen species; NADPH, nicotinamide adenine dinucleotide phosphate; VDAC, voltage-dependent anion channel; TGF-β1, transforming growth factor-β 1; PG, prostaglandin; PTGS2, prostaglandin-endoperoxide synthase 2; LOX, lipoxygenase; LPCAT3, phospholipid choline acyltransferase 3; AA(PUFAs), arachidonic acid (Polyunsaturated fatty acids); AA-CoA, arachidonic acid-coenzyme A; ACSL4, acyl-Coa synthase long-chain family member 4; NF2, neurofibromatosis 2; YAP, Yes-associated protein; MAPK, mitogen-activated protein kinase; TNF-α, tumor Necrosis Factor-α; IL, interleukin; Smad3, Small mothers against decapentaplegic.

**Figure 2 cells-11-03349-f002:**
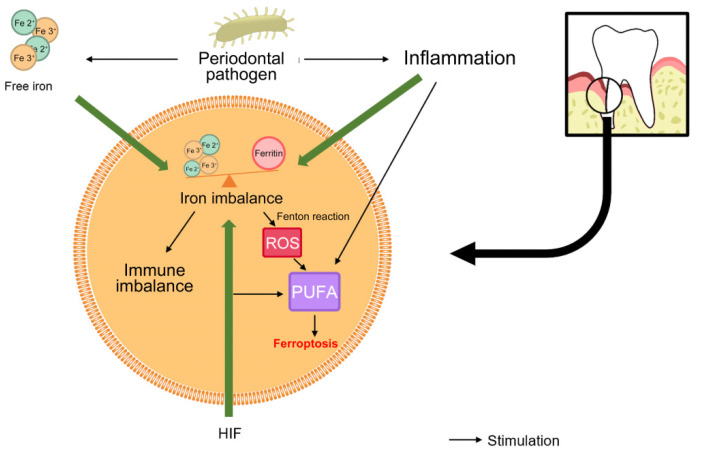
**Ferroptosis programs in the periodontal microenvironment.** In the periodontitis environment, inflammation, HIF, and iron concentration influence intracellular iron homeostasis. Intracellular iron homeostasis is responsible for immune homeostasis. In an inflammatory state, periodontal pathogenic micro-organisms up-regulate iron ion concentration through metabolism. At the same time, inflammation and HIF promote the deposition of PUFA in cells. When intracellular iron homeostasis is unbalanced, the Fenton reaction will occur and reactive oxygen species will be generated to peroxidation of intracellular PUFA and generate lipid peroxides to accelerate the process of cell ferroptosis. Abbreviations—ROS, reactive oxygen species; PUFA, polyunsaturated fatty acids; HIF, hypoxia-inducible factor.

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
