# Peer review of "Ferroptosis: A New Development Trend in Periodontitis"

_cells, 2022, doi:10.3390/cells11213349_

Round 1

Reviewer 1 Report

This review summarizes data on the relationship between ferroptosis and periodontitis. It is well organized,  addressing various aspects related to both ferroptosis and periodontitis. It is written quite well, also from the linguistic point of view, few sentences need to be reviewed as a grammatical construction. The only minor point concerns the figures: the caption of Figure A does not seem well written to me, it is a bit confusing and thus it should be rewritten; the font size within the design should be slightly increased. The blank letters within Figure B are difficult to read.

Author Response

Response to Reviewer 1 Comments

Point 1: This review summarizes data on the relationship between ferroptosis and periodontitis. It is well organized, addressing various aspects related to both ferroptosis and periodontitis. It is written quite well, also from the linguistic point of view, few sentences need to be reviewed as a grammatical construction. The only minor point concerns the figures: the caption of Figure A does not seem well written to me, it is a bit confusing and thus it should be rewritten; the font size within the design should be slightly increased. The blank letters within Figure B are difficult to read.

Response 1: We really appreciate your opinion on the importance of our study and your comments on our paper that help improve its quality. We have combed and corrected the title of Figure A in the manuscript. Please see the annotated section of Figure A in the revised manuscript for details. In addition, we make appropriate modifications to Figure A and Figure B, including details such as color and font, which is expected to be helpful for reading and understanding the article.

Reviewer 2 Report

-          The main issue about this paper is how the authors organize this review. They try to systematically review the related mechanisms of ferroptosis. So, a discussion paragraph should not be written. All these items could be part of the previous paragraphs. The discussion is the comparison of the results with the existing data.

-          When the authors describe the ferroptosis in periodontitis they should cited the papers in this specific field, avoiding the description of ferroptosis in other general systemic conditions, these one should be part of previous paragraphs  

Author Response

Response to Reviewer 2 Comments

Point 1: The main issue about this paper is how the authors organize this review. They try to systematically review the related mechanisms of ferroptosis. So, a discussion paragraph should not be written. All these items could be part of the previous paragraphs. The discussion is the comparison of the results with the existing data.

Response 1: Thanks for this constructive suggestion. We have integrated some of the contents of the discussion into the previous relevant paragraphs, and modified the original "discussion" to "clinical applications and conclusions" to explain the existing research applications and conclusions about the topic. Please see section “CLINICAL APPLICATIONS AND CONCLUSSION” in the revised manuscript.

Point 2: When the authors describe the ferroptosis in periodontitis they should cited the papers in this specific field, avoiding the description of ferroptosis in other general systemic conditions, these one should be part of previous paragraphs.

Response 2: Thank you very much for your kind assessment and the useful comments to improve our manuscript. We extract the description of ferroptosis in other general systemic conditions and merge into the previous sections, and focus on the periodontal descriptions in the “Ferroptosis and periodontitis” section. Please see section “Lipid peroxidation”, “MAPK pathway“, and “P53 pathway” in the revised manuscript. The structure of the revised manuscript is really conducive to understanding. We therefore thank you for your helpful comments and comments.

Reviewer 3 Report

Reviewer Comments:

Manuscript titled “Ferroptosis: a new development trend in periodontitis.”

The review article is interesting as the authors, Chen K et al. have emphasized on the relationship between periodontitis and ferroptosis, to provide a theoretical reference for future research on the prevention and treatment of periodontal disease. Though the article is well written, it has number of errors/flaws in the text as mentioned below.

1.     Periodontitis is an inflammatory disease; hence, the authors should elaborate more on inflammation and the role of microorganisms in periodontitis in context to ferroptosis. The authors have covered some aspects under header “ferroptosis and periodontitis”, however, is limited.

2.     The registered trademark sign “R” is repeated number of times which is meaningless – line 28, 149, 150, 249, 298, 299, 355 and so on… throughout the text.

3.     The line 36 - “Fe” abbreviation – write the full form when appear for first time.

4.     Some of the key references on Ferroptosis from years 2021 and 2022 are missing – should be incorporated.

5.     Major errors in Figure A – improvise and revise the figure

a.     Mention full forms of all abbreviations used in the figure at the end of the legend.

b.   Cystine arrow is pointing outward which should be inward – into the cell as glutamate is transferred out of the cell and cystine vice versa.

c.   The authors should elaborate on the role of other subunit Slc3a2 in manuscript – as its function is not mentioned throughout the manuscript text

d.   The authors failed to show some aspects that they mentioned in the legends such as, “Ferritin autophagy occurs through NCOA4” – no NCOA4 in figure – NCOA4 is an important ferroptosis related gene in periodontitis; “P53 not only inhibits the expression of SLC7A11” – no inhibition sign shown; MAPK pathway and hippo pathway promote each other, NF2 activates hippo, activates LATS1/2, phosphorylates YAP and inhibits the expression of downstream ACSL4 – no mention of LATS1/2 in the figure.

6.     Major errors in Figure B – improvise and revise the figure

a.   Mention full forms of all abbreviations used in the figure at the end of the legend.

b.   Figure shows iron homeostasis but in fact, ferroptosis occurs during iron imbalance which is not reflected in the figure – correct the error

c.   It should be “inflammation” instead of the word “immflamation”

7.     Spell check for errors throughout the manuscript.

8.     Reference section has lots of errors, e.g reference 3; and in some journal names are abbreviated and some has full form – amend as per journal format

9.     Improvised the manuscript text as per the suggestions.

Author Response

Response to Reviewer 3 Comments

Point 1: Periodontitis is an inflammatory disease; hence, the authors should elaborate more on inflammation and the role of microorganisms in periodontitis in context to ferroptosis. The authors have covered some aspects under header “ferroptosis and periodontitis”, however, is limited.

Response 1: Thank you very much for your kind assessment and the useful comments to improve our manuscript. We added a cyclic feedback of the microbial-iron-dependent oxidative stress-inflammation triad to periodontitis and explained the importance of lipid peroxidation in the context of ferroptosis and periodontitis. We hope this will make our explanations clearer.

Point 2: The registered trademark sign “R” is repeated number of times which is meaningless – line 28, 149, 150, 249, 298, 299, 355 and so on… throughout the text.

Response 2: This suggestion is appreciated. To this end, we have deleted the sign “R” in the revised manuscript.

Point 3: The line 36 - “Fe” abbreviation – write the full form when appear for first time.

Response 3: We agree with your opinion that is kind and worthy of our appreciation. We have added this in the second paragraph of the revised manuscript.

Point 4: Some of the key references on Ferroptosis from years 2021 and 2022 are missing – should be incorporated.

Response 4: We really appreciate your opinion on the importance of our study and your comments on our paper that help improve its quality. We realized that there was a real shortage of references on ferroptosis since 2020, so we added recent references as appropriate. We hope that this will help readers to keep abreast of the latest research progress.

Point 5: Major errors in Figure A – improvise and revise the figure

  1. Mention full forms of all abbreviations used in the figure at the end of the legend.
  2. Cystine arrow is pointing outward which should be inward – into the cell as glutamate is transferred out of the cell and cystine vice versa.
  3. The authors should elaborate on the role of other subunit Slc3a2 in manuscript – as its function is not mentioned throughout the manuscript text
  4. The authors failed to show some aspects that they mentioned in the legends such as, “Ferritin autophagy occurs through NCOA4” – no NCOA4 in figure – NCOA4 is an important ferroptosis related gene in periodontitis; “P53 not only inhibits the expression of SLC7A11” – no inhibition sign shown; MAPK pathway and hippo pathway promote each other, NF2 activates hippo, activates LATS1/2, phosphorylates YAP and inhibits the expression of downstream ACSL4 – no mention of LATS1/2 in the figure.

Response 5: Thank you for your kindness and patience to point out these details. In the newly submitted figure A, we have added and modified the LATS1/2, inhibition of P53 to SLC7A11, arrow direction, abbreviation and other issues you mentioned. We checked NCOA4, which is in this picture. As for Slc3a2, in the process of ferroptosis, it is more involved in the subunit structure of System Xc-, so we did not elaborate in detail. Finally, we would like to thank you again for your guiding comments, which have benefited us greatly.

Point 6: Major errors in Figure B – improvise and revise the figure

  1. Mention full forms of all abbreviations used in the figure at the end of the legend.
  2. Figure shows iron homeostasis but in fact, ferroptosis occurs during iron imbalance which is not reflected in the figure – correct the error
  3. It should be “inflammation” instead of the word “immflamation”

Response 6: Thank you for your kindness and patience to point out these details. We have corrected and revised the icons, abbreviations, and words you pointed out. Please see the newly submitted figure B.

Point 7: Spell check for errors throughout the manuscript.

Response 7: Thank you very much for your kind assessment and the useful comments to improve our manuscript. We have carefully proofread and verified the manuscript to eliminate spelling errors.

Point 8: Reference section has lots of errors, e.g reference 3; and in some journal names are abbreviated and some has full form – amend as per journal format.

Response 8: Thank you very much for your kind assessment and the useful comments to improve our manuscript. We have made modifications based on this comment. Please see section “References” in the revised manuscript.

Point 9: Improvised the manuscript text as per the suggestions.

Response 9: Thank you very much for your kind assessment and the useful comments to improve our manuscript. We hope that the changes made are consistent with the reviewer’s expectations.

Reviewer 4 Report

the paper is interesting. nevertheless, the methodology of systematic review is missing. I suggest to reorganize the article and resubmit

Author Response

Response to Reviewer 4 Comments

Point 1: The paper is interesting. nevertheless, the methodology of systematic review is missing. I suggest to reorganize the article and resubmit

Response 1: Your comments have great guiding significance for me and my future writing. We acknowledge that this manuscript is a review, not a systematic review. Nonetheless, given the understanding of the mechanisms and clinical applications of ferroptosis in periodontitis, we believe that writing this article could well facilitate future studies. We truly believe that your comments can be a constructive guide to our articles and work, so we have supplemented the article with our search methods and the details of the structure of the article. Please refer to the fourth paragraph of the revised manuscript. Finally, we would like to thank you again for your guiding comments, which have benefited us greatly.

Round 2

Reviewer 2 Report

Now it can be accepted

Author Response

Response: Thank you for your suggestions.

Reviewer 3 Report

Most of the comments have been addressed by the authors. The authors need to double check the contribution part.

Author Response

Point: The authors need to double check the contribution part.

Response: Thanks for this kind suggestion. We have again taken into account the contributions of the authors and have included a note on the changes made in the manuscript, together with a table of author contributions.

Reviewer 4 Report

I still think that this a simple review regarding the journal, but the authors have made an evident effort to improve the paper. if the editor agrrees if could be consider to be published in the journal

Author Response

Point: I still think that this a simple review regarding the journal, but the authors have made an evident effort to improve the paper. if the editor agrrees if could be consider to be published in the journal.

Response: Thank you for your suggestion. This will be the direction our future research should strive for.